# Growth and Mineral Relations of *Beta vulgaris* var. *cicla* and *Beta vulgaris* ssp. *maritima* Cultivated Hydroponically with Diluted Seawater and Low Nitrogen Level in the Nutrient Solution

**Martina Puccinelli** [1,*], **Giulia Carmassi** [1], **Luca Botrini** [1], **Antonio Bindi** [1], **Lorenzo Rossi** [2], **Juan Francisco Fierro-Sañudo** [1], **Alberto Pardossi** [1] **and Luca Incrocci** [1]

[1] Department of Agriculture, Food and Environment, University of Pisa, Via del Borghetto 80, 56124 Pisa, Italy; giulia.carmassi@unipi.it (G.C.); luca.botrini@unipi.it (L.B.); a.bindi3@studenti.unipi.it (A.B.); jf.fierro26@gmail.com (J.F.F.-S.); alberto.pardossi@unipi.it (A.P.); luca.incrocci@unipi.it (L.I.)

[2] Department of Veterinary Sciences, University of Pisa, Viale delle Piagge 2, 56124 Pisa, Italy; lorenzo.rossi@phd.unipi.it

\* Correspondence: martina.puccinelli@agr.unipi.it

**Abstract:** There is an increasing interest in the use of seawater in horticulture. The objective of this study was to evaluate *Beta vulgaris* var. *cicla* (Swiss chard) and its wild ancestor *B. vulgaris* spp. *maritima* (sea beet) as potential crop species for seawater hydroponics or aquaponics. Both species were grown in a floating system for leaf production with recurrent harvests. The nutrient solutions contained different concentrations of nitrate (1 and 10 mM) and a synthetic sea salt (0 and 10 g L$^{-1}$), in a factorial design, where the saline solution with a low nitrate level intended to mimic the typical nutritional conditions of saltwater aquaponics. In both species, increasing the salinity or reducing the N level in the nutrient solution reduced the crop yield and total dry biomass. In both Swiss chard and sea beet, the use of saline water resulted in a lower leaf concentration of K, Ca, Cu, and Mn, and a greater content of Na and Cl. In Swiss chard, an increase in Na and Cl and a decrease in K leaf content were found in successive harvests. On average, sea beet showed a higher leaf production and accumulation of nitrate than Swiss chard.

**Keywords:** 'cut and come again' harvest; floating system; halophytes; leafy vegetables; sea beet; Swiss chard

## 1. Introduction

As freshwater becomes limited, brackish to highly saline waters, including seawater, used after desalinization or dilution with freshwater, are alternative resources for crop irrigation [1,2]. A particular example of seawater application for crop production is saltwater aquaponics (or haloponics), which uses water with a wide range of salinities, up to 35 g L$^{-1}$ (the salinity of seawater) [3]. In aquaponics, crop production can be limited by sub-optimal and irregular concentrations of nutrients. For instance, the typical N concentration in aquaculture wastewater is 0.1 to 2 mM ([4], and references cited therein) and thus is much lower compared to the optimum N level (6–16 mM) in hydroponic culture solutions [5]. Additionally, in seawater aquaponics, plant growth can be negatively affected by high water salinity and the concentration of some nutrients, such as magnesium (Mg), boron (B), sodium (Na), and chloride (Cl), and by unusual molar ratios between nutrients (e.g., Ca/Mg) [6,7].

The possibility of using seawater for crop irrigation and hydroponic production has awakened new interest in plant species with inherent salt tolerance [1]. The *Beta* genus, in the *Amaranthaceae* family, includes several salt-tolerant species that could be cultivated

using saline water, such as *Beta vulgaris* L. var. *cicla* (also known as Swiss chard, SC) and *Beta vulgaris* L. ssp. *maritima* (also known as sea or wild beet, SB).

Swiss chard is sensitive to salinity, especially at germination and seedling stage [8], but it is quite tolerant at adult stage [9]. Swiss chard is a popular leafy vegetable around the world, which it is mainly used cooked, as a side dish or in soups [10]. Sea beet is a wild ancestor of all beet crops, which grows naturally in coastal areas, salt marsh, and saline regions in the Mediterranean area and in northern Europe [10]. Its leaves are usually eaten cooked [11]. Sea beet is a facultative halophyte that is more tolerant to drought and salt stress than the cultivated varieties of *B. vulgaris*, including SC [12]. In previous works that were conducted with plants grown hydroponically with different NaCl concentrations in the nutrient solution, plant growth was reduced above 4.68 and 7.31 g $L^{-1}$, respectively, in SC [13] and SB [14].

Several studies have been published on SC that is cultivated in hydroponics (e.g., [15,16]; see also references cited therein) or aquaponics (e.g., [6,7,17]; references cited therein). Very few works, on the other hand, have been conducted on SB that is grown in hydroponics [18] and aquaponics [19].

The main objective of this study was to evaluate SC and SB as potential crop species for hydroponic or aquaponic cultivation with diluted seawater. Therefore, both species were grown hydroponically (in a floating system) using freshwater or diluted artificial seawater (10 g $L^{-1}$) and two concentrations of nitrate nitrogen (N-NO$_3$; 1 and 10 mM) in the nutrient solution, in a factorial design, where the non-saline solution with 10 mM of N-NO$_3^-$ was the control and the saline solution with 1 mM of N-NO$_3^-$ intended to mimic the typical nutritional conditions of saltwater aquaponics. This N concentration was chosen since it is close to the value of the typical N concentration in aquaculture wastewater [20]. Moreover, high-value euryhaline fish species, such as European sea bass (*Dicentrarchus labrax* L.) and Gilt-head Sea bream (*Sparus aurata* L.), could be reared at the salinity level (10 g $L^{-1}$) that was tested in the present work [21].

The plants were cultivated at a high plant density with multiple harvests [22]. Therefore, another goal of this work was to investigate the effects of successive cuts ('cut and come again' harvest) on some leaf quality attributes that were associated with mineral content. Very few studies have been published on the effect of recurrent cuts on leaf production and the mineral relations of SC that is grown in hydroponics [16] or aquaponics [6], and we are not aware of any work that is conducted with SB.

## 2. Materials and Methods

### 2.1. Plant Material and Growing Conditions

The experiment was conducted in a glasshouse at the University of Pisa, Italy (lat. 43°42′42″48 N, long. 10°24′52″92 E), between late winter and spring 2020, under natural light. The climatic conditions were continuously monitored by a weather station that was located inside the greenhouse. Basic information on the experiment is reported in Table 1.

**Table 1.** Basic information on the experiment with Swiss chard and sea beet grown hydroponically under greenhouse.

| Sowing | | 17 February 2020 | |
|---|---|---|---|
| Transplant | | 9 March 2020 | |
| Start of treatment | | 23 March 2020 (14 DAT *) | |
| | 1st cut (C1) | 2nd cut (C2) | 3rd cut (C3) |
| Harvest date | 6 April 2020 | 20 April 2020 | 4 May 2020 |
| Days of treatment | 14 (28 DAT) | 14 (42 DAT) | 14 (56 DAT) * |
| Mean air temperature (°C) | 21.6 | 23.2 | 23.1 |
| Mean daily solar radiation (MJ m$^{-2}$ day$^{-1}$) | 10.0 ** | 12.8 | 12.7 |
| Cumulative solar radiation (MJ m$^{-2}$) | 291.0 ** | 179.2 | 177.2 |

* DAT stands for the number of days after transplanting. ** The values were computed for the period from transplanting to the first cut.

Seeds of SC and SB were purchased, respectively, from Gargini Sementi (Lucca, Italy; https://www.garginisementi.it, accessed on 3 February 2020) and from Pennard Plants (Shepton Mallet, UK, https://www.pennardplants.com/, accessed on 3 February 2020), and sown in 180-cell trays with rockwool plugs. The trays were placed in a growth chamber at 25 °C for five days, and the seedlings were planted in about 50-L plastic tanks (water depth 25 cm) with a stagnant nutrient solution 21 days after sowing. Each tank hosted 180 plants, and there were four tanks per m$^2$; therefore, the crop density was approximately 720 plants per m$^{-2}$ of ground area.

Diluted seawater was prepared using the synthetic sea salt Instant Ocean$^{TM}$ (IO, Askoll Uno, Sandrigo, Vicenza, Italy), which is widely used for marine aquaria and scientific research [23]. Four nutrient solutions with different concentrations of IO (0 and 10 g L$^{-1}$) and N-NO$_3^-$ (1 and 10 mM) were tested in a completely randomized design with three replicates for each treatment; a replicate consisted of one tank with 180 plants. The solution without IO and containing 10 mM of N-NO$_3^-$ was the control. The IO content of macronutrients and trace elements, which was determined by an external laboratory (Demetra snc, Pescia, Italy) using UV/VIS spectrophotometry, atomic absorption spectroscopy (AAS) or inductively coupled plasma (ICP), was the following: P 2.5 mg kg$^{-1}$; K 8514 mg kg$^{-1}$; Ca 8914 mg kg$^{-1}$; Mg 28,857 mg kg$^{-1}$; Na 259,413 mg kg$^{-1}$; Cl 525,714 mg kg$^{-1}$; S-SO$_4$ 23,440 mg kg$^{-1}$; B 113.9 mg kg$^{-1}$; Fe 17.1 mg kg$^{-1}$; Mn 2.6 mg kg$^{-1}$; Zn 5.65 mg kg$^{-1}$; Cu 1.71 mg kg$^{-1}$. The content of N-NO$_3$, N-NH$_4$ and molybdenum was below the detection limit.

The ion concentrations, pH, and EC of the four nutrient solutions are reported in Table 2, along with their abbreviations used in the text. The level of ammonium and nitrite were below the detection limits. The difference between the content of S-SO$_4$ in 0IO-10N and 0IO-1N was due to the fertilizers that were used. In 0IO-1N, a higher amount of K$_2$SO$_4$ was used instead of KNO$_3$ to provide the desired amount of K. Likewise, the different content of Mg in 10IO-10N and 10IO-1N was due to the different amount of Mg (NO$_3$)$_2$ that was used to reach the desired N-NO$_3$ level.

**Table 2.** Mineral composition, electrical conductivity (EC), and pH of the four nutrient solutions used in the experiment with Swiss chard and sea beet grown hydroponically under greenhouse.

| | Treatment Abbreviation | | | |
| --- | --- | --- | --- | --- |
| | 0IO-10N | 0IO-1N | 10IO-10N | 10IO-1N |
| Instant Ocean salt (IO; g L$^{-1}$) | 0 | 0 | 10.00 | 10.00 |
| N-NO$_3$ (mM) | 10.00 | 1.00 | 10.00 | 1.00 |
| P (mM) | 0.50 | 0.50 | 0.50 | 0.50 |
| K (mM) | 9.00 | 9.00 | 9.00 | 9.00 |
| Ca (mM) | 4.50 | 4.50 | 4.50 | 4.50 |
| Mg (mM) | 2.00 | 2.00 | 17.48 | 12.98 |
| Na (mM) | 8.58 | 8.58 | 110.80 | 110.80 |
| S-SO$_4$ (mM) | 1.28 | 5.78 | 11.21 | 11.21 |
| Cl (mM) | 14.87 | 14.87 | 146.40 | 146.40 |
| Fe (µM) | 40.00 | 40.00 | 40.00 | 40.00 |
| B (µM) | 40.00 | 40.00 | 103.74 | 103.74 |
| Cu (µM) | 3.00 | 3.00 | 145.73 | 145.73 |
| Zn (µM) | 10.00 | 10.00 | 10.00 | 10.00 |
| Mn (µM) | 10.00 | 10.00 | 10.00 | 10.00 |
| Mo (µM) | 1.00 | 1.00 | 1.00 | 1.00 |
| Electrical conductivity (dS m$^{-1}$) | 3.10 | 3.06 | 14.52 | 14.82 |
| pH | 5.60 | 5.60 | 5.60 | 5.60 |

The nutrient solutions were prepared using groundwater and appropriate amounts of IO and inorganic salts of technical grade. To avoid osmotic shock, half of the final IO content in each tank was dissolved nine days after planting and then five days later.

In all the tanks, the nutrient solution was continuously aerated, and the dissolved oxygen remained above 6 mg $L^{-1}$ during the whole experiment. When the level of water in each tank decreased by approximately 10% because of plant water uptake, the tank was refilled with newly prepared nutrient solution. The pH and N-$NO_3^-$ level in the nutrient solution were measured every 1–3 days using a handy pH-meter and a colorimetric assay kit (Nitrate Cuvette Test Spectroquant, Merck Life Science, Milano, Italy); the pH was adjusted to 6.0–6.5 and the N-$NO_3^-$ concentration to 1.0 or 10 mM, when necessary. The nutrient solution was completely renewed every week.

In both species, leaves were harvested trice at 28, 42, and 56 days after transplanting (Table 1); leaves were cut approximately 2 cm above the base of the plant.

### 2.2. Determinations

#### 2.2.1. Crop Yield, Plant Growth and Water Uptake

The total crop yield was determined by recording the fresh leaf biomass of all the plants in each hydroponic tank in successive harvests. The leaf dry weight (DW) was determined after drying in a ventilated oven at 70 °C until reaching constant weight. At each harvest, one sample consisting of three individual plants was collected from each replicate tank to determine the leaf area and succulence. The leaf area was measured using an electronic planimeter (DT Area Meter MK2, Delta T-Devices) and leaf succulence was calculated as the ratio between leaf fresh weight (FW) and area. The root DW was also determined at the end of the experiment.

The plant water uptake was estimated by measuring the water that was used to refill the hydroponic tanks throughout the experiment; the water surface was completely covered by the floating trays and therefore the amount of water that was lost through evaporation was negligible.

#### 2.2.2. Plant Mineral Content and Uptake

Dry samples of the leaves or roots were digested with a mixture (5:2) of nitric acid (65%) and perchloric acid (35%) at 240 °C for 1 h, and elements were determined as follows: Ca, Mg, Cu, Fe, Mn, and Zn by atomic absorption spectroscopy; P by UV/VIS spectrometry (Olsen's method); K and Na by flame photometry. The nitrate ($NO_3^-$) content was measured spectrophotometrically in dry leaf samples that were extracted with distilled water (100 mg DW in 20 mL) at room temperature for 2 h, using the salicylic-sulfuric acid method. The same extract was used to analyze the content of Cl by ion chromatography.

The total plant uptake of mineral nutrients, Na and Cl was calculated based on the dry weight and mineral content measured in the leaves of each harvest and in the roots at the end of the experiment. The daily rate of N uptake was calculated by dividing the N content of the leaves of each harvest by the number of days between the transplanting and the first cut (28 days), and between the successive cuts (14 days).

### 2.3. Contribution of Leaf Consumption to Mineral Dietary Intake and Health Risk

The estimated daily intake (EDI, mg day$^{-1}$) of mineral elements resulting from the consumption of leaves of SW or SB was calculated using the mineral content of fresh leaves and considering a daily serving size of 100 g of fresh leaves for an average adult. The values of EDI were expressed as a percentage of the reference intake (RI) for adults, which are reported for P, K, Ca, Mg Cl, Cu, Mn, Fe and Zn in the Annex XIII of the Regulation (EU) No. 1169/2011 [24].

The health risk index (HRI) due to excess intake of minerals was also calculated as the ratio between EDI and the tolerable upper intake level (UIL; mg day$^{-1}$) for Ca, Cu, Mg, and Zn, the safe and adequate intake (SAI) for Cl and Na, or the acceptable daily intake (ADI; for a 60 kg human) for $NO_3$ (Table S1).

*2.4. Statistical Analysis*

The data were tested for homogeneity of variances using Levene's test. The data of total biomass production and mineral uptake were subjected to a 3-way ANOVA with salinity and N-NO$_3$ concentration of the nutrient solution, and plant species as variables. The effects of cut time on leaf production and mineral content was subjected to a 4-way nested ANOVA with the salinity and N-NO$_3$ concentration of the nutrient solution, plant species, and cut as variables (Table S2). As the interaction 'salinity × N level × cut × species' was not significant for most of the measured parameters, for the sake of simplicity, the effect of cut was separately analyzed in SC and SB.

The mean values were separated by Tukey's posthoc test or a *t*-test (crop yield) ($p < 0.05$). The statistical analysis was performed using R Statistical Software.

## 3. Results

*3.1. Crop Yield and Mineral Relations*

3.1.1. Crop Yield

The plant species, salinity and N-NO$_3$ level in the nutrient solution significantly influenced plant growth and total crop yield, but the interactions among these factors were not significant in most cases (Figure 1; Table 3). On average, the crop yield was significantly lower in SC (−16%) than in SB and, compared to the control (0IO-10N), when the plants were grown in diluted seawater (−30%) or with lower N-NO$_3$ level in the solution (−25%; Table 3).

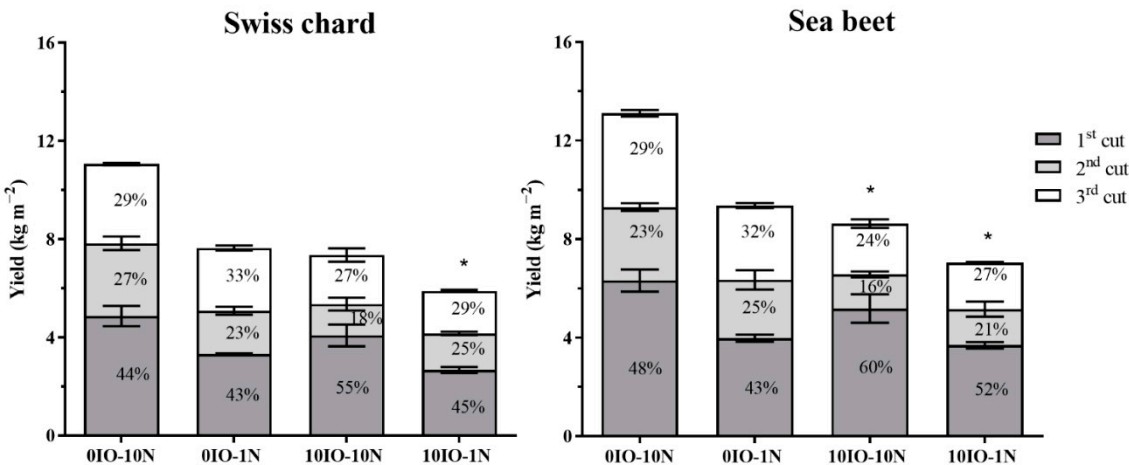

**Figure 1.** Crop yield (fresh leaves) of Swiss chard (left) and sea beet (right) grown hydroponically under greenhouse with different concentrations of the sea salt Instant Ocean (IO; 0 and 10 g L$^{-1}$) and nitrogen (1 and 10 N-NO$_3$ mM) in the nutrient solution (see Table 2 for abbreviations). Plants were harvested trice during the experiment; the values inside the bar are the percent contribution of each harvest to the total crop yield. Mean values (±s.e.; *n* = 3) keyed by the asterisk (*) are statistically different from the control (0IO-10N), according to the *t*-test.

The contribution of the first harvest to the total leaf production was greater (43% to 60%, depending on the treatment) as compared with the second (16% to 29%,) and third cut (24% to 34%), because of the longer growing period between the transplant and the first cut than between the successive cuts (28 days against 14 days; Figure 1). Similar results were found as regards the total leaf DW, which was not significantly influenced by the salinity level (Table 3). Leaf DW was lower (−32%) in SC than in SB, and in plants that were grown with 1 mM of N-NO$_3$ (−14%, on average). In SC, the high salinity decreased the root DW (−42%). The root/leaf DW ratio (−29%) was much smaller (−55%) in SB than in SC (Table 3).

**Table 3.** Total crop yield (leaf fresh weight, FW) and dry weight (DW) of leaves, roots and whole plants, root/leaf DW ratio, and total uptake of water and nitrogen (N) of Swiss chard and sea beet plants grown hydroponically with different concentrations of the sea salt Instant Ocean (IO; 0 and 10 g L$^{-1}$) and nitrogen (1 and 10 N-NO$_3$ mM) in the nutrient solution (see Table 2 for abbreviations). in the nutrient solution.

| Plant Species | IO (g L$^{-1}$) | N-NO$_3$ (mM) | Yield (kg FW m$^{-2}$) | Leaf DW (kg m$^{-2}$) | Root DW (kg m$^{-2}$) | Total DW (kg m$^{-2}$) | Root/Leaf (%) | Water Uptake (L m$^{-2}$) |
|---|---|---|---|---|---|---|---|---|
| Swiss chard | 0 | 10 | 11.06 | 0.470 | 0.086 | 0.543 | 15.4 | 217.3 a |
| | | 1 | 7.64 | 0.424 | 0.093 | 0.531 | 25.2 | 208.5 ab |
| | 10 | 10 | 7.35 | 0.448 | 0.050 | 0.499 | 11.4 | 156.3 cd |
| | | 1 | 5.89 | 0.400 | 0.055 | 0.461 | 15.3 | 160.1 cd |
| Sea beet | 0 | 10 | 13.11 | 0.768 | 0.101 | 0.851 | 10.8 | 178.7 bc |
| | | 1 | 9.36 | 0.632 | 0.045 | 0.668 | 5.8 | 161.4 cd |
| | 10 | 10 | 8.63 | 0.639 | 0.065 | 0.691 | 8.0 | 135.3 d |
| | | 1 | 7.05 | 0.547 | 0.044 | 0.579 | 5.9 | 131.5 d |
| MEAN EFFECTS | | | | | | | | |
| Swiss chard | | | 7.99 b | 0.436 b | 0.072 | 0.508 b | 16.8 a | 185.5 a |
| Sea beet | | | 9.54 a | 0.646 a | 0.064 | 0.697 a | 7.6 b | 151.7 b |
| | 0 | | 10.29 a | 0.573 | 0.082 a | 0.648 a | 14.3 a | 191.5 a |
| | 10 | | 7.23 b | 0.509 | 0.053 b | 0.557 b | 10.1 b | 145.8 b |
| | | 10 | 10.04 a | 0.581 a | 0.074 | 0.646 a | 11.4 | 171.9 |
| | | 1 | 7.49 b | 0.501 b | 0.062 | 0.560 b | 13.0 | 165.4 |
| Swiss chard | 0 | | 9.35 | 0.447 | 0.089 a | 0.537 | 20.3 a | 212.9 a |
| | 10 | | 6.62 | 0.424 | 0.052 b | 0.480 | 13.3 ab | 158.2 b |
| Sea beet | 0 | | 11.24 | 0.700 | 0.073 ab | 0.759 | 8.3 bc | 170.1 b |
| | 10 | | 7.84 | 0.593 | 0.054 b | 0.635 | 7.0 c | 133.4 c |
| Swiss chard | | 10 | 9.21 | 0.459 | 0.068 | 0.521 | 13.4 b | 186.8 |
| | | 1 | 6.76 | 0.412 | 0.077 | 0.496 | 20.2 a | 184.3 |
| Sea beet | | 10 | 10.87 | 0.703 | 0.083 | 0.771 | 9.4 bc | 157.0 |
| | | 1 | 8.21 | 0.589 | 0.044 | 0.623 | 5.8 c | 146.5 |
| | 0 | 10 | 12.09 a | 0.619 | 0.092 | 0.697 | 13.1 | 198.0 |
| | | 1 | 8.50 b | 0.528 | 0.072 | 0.599 | 15.5 | 185.0 |
| | 10 | 10 | 7.99 bc | 0.544 | 0.056 | 0.595 | 9.7 | 145.8 |
| | | 1 | 6.47 c | 0.474 | 0.049 | 0.520 | 10.6 | 145.8 |
| ANOVA | | | | | | | | |
| Plant species (PS) | | | ** | *** | ns | *** | *** | * |
| IO concentration | | | *** | ns | ** | * | ** | *** |
| N-NO$_3$ concentration | | | *** | * | ns | * | ns | ns |
| PS × IO | | | ns | ns | ns | ns | * | *** |
| PS × N-NO$_3$ | | | ns | ns | * | ns | ** | ns |
| IO × N-NO$_3$ | | | * | ns | ns | ns | ns | ns |
| PS × IO × N-NO$_3$ | | | ns | ns | ns | ns | ns | ** |

Means (*n* = 3) keyed by the same letter are not statistically different for *p* = 0.05 after Duncan's test. Significance level: *** $p \leq 0.001$; ** $p \leq 0.01$; * $p \leq 0.05$; ns = not significant.

### 3.1.2. Root Mineral Content

The root N content was neither significantly affected by the plant species nor by the salinity and N concentration of the nutrient solution (Tables S3 and S4).

The concentration of P, K, Na, and Cl, and the K/Na molar ratio were similar in SB and SC (Table S3). On average, the saline nutrient solution increased the root content of Na (+234%) and Cl (+72%), whereas it reduced the P concentration (−44.5%) and the K/Na molar ratio (−68%). A low N level increased the root content of K in SB (+29%; Table S3).

The concentrations of trace elements in root tissues were similar in SC and SB, and there were no important effects of nutrient solution salinity and N level (data not shown).

### 3.1.3. Water and Mineral Uptake

On average, the water uptake was significantly lower in SB (−18%) than in SC, and in salinized plants than in non-salinized plants (−24%) (Table 3).

The total uptake of N (+59%) was larger in SB than in SC and, as expected, in plants that were grown at the higher N concentration (+50%; Table 4). The total P uptake did

not differ in SB and SC, and on average it was reduced (−40%) by the salinization of the nutrient solution, while in SC it increased (+53%) at a reduced N level (Table 4).

**Table 4.** Total uptake of nitrogen (N); phosphorus (P); potassium (K); sodium (Na); calcium (Ca); magnesium (Mg); and chloride (Cl) in Swiss chard and sea beet plants grown hydroponically with different concentrations of the sea salt Instant Ocean (IO; 0 and 10 g L$^{-1}$) and nitrogen (1 and 10 N-NO$_3$ mM) in the nutrient solution.

| Plant Species | IO (g L$^{-1}$) | N-NO$_3$ (mM) | N | P | K | Ca (g m$^{-2}$) | Mg | Cl | Na |
|---|---|---|---|---|---|---|---|---|---|
| Swiss chard | 0 | 10 | 25.72 | 1.14 cd | 40.77 | 4.12 | 4.22 cd | 20.12 | 16.76 |
| | | 1 | 19.76 | 2.34 a | 33.92 | 3.79 | 4.84 c | 22.44 | 15.75 |
| | 10 | 10 | 24.42 | 1.00 cd | 33.52 | 2.34 | 4.54 c | 24.43 | 25.17 |
| | | 1 | 16.61 | 0.94 cd | 20.46 | 2.09 | 2.94 d | 27.08 | 26.24 |
| Sea beet | 0 | 10 | 45.79 | 1.61 b | 77.66 | 6.60 | 9.05 a | 25.52 | 38.95 |
| | | 1 | 27.86 | 1.29 bc | 45.01 | 5.25 | 4.69 c | 21.38 | 46.18 |
| | 10 | 10 | 38.78 | 1.06 cd | 49.09 | 3.70 | 7.16 b | 24.97 | 55.32 |
| | | 1 | 25.32 | 0.83 d | 36.04 | 3.66 | 4.40 cd | 23.55 | 54.87 |
| MEAN EFFECTS | | | | | | | | | |
| Swiss chard | | | 21.6 b | 1.36 | 32.17 b | 3.09 b | 4.13 b | 23.52 | 20.98 b |
| Sea beet | | | 34.4 a | 1.20 | 51.95 a | 4.80 a | 6.32 a | 23.86 | 48.83 a |
| | 0 | | 29.78 | 1.60 a | 49.34 a | 4.94 a | 5.70 a | 22.4 | 29.41 b |
| | 10 | | 26.28 | 0.96 b | 34.78 b | 2.95 b | 4.76 b | 25.0 | 40.40 a |
| | | 10 | 33.68 a | 1.20 | 50.26 a | 4.19 | 6.24 a | 23.8 | 34.05 |
| | | 1 | 22.39 b | 1.35 | 33.86 b | 3.70 | 4.22 b | 23.6 | 35.76 |
| Swiss chard | 0 | | 22.74 | 1.74 | 37.34 | 3.96 | 4.53 | 21.28 | 16.26 |
| | 10 | | 20.52 | 0.97 | 26.99 | 2.21 | 3.74 | 25.75 | 25.71 |
| Sea beet | 0 | | 36.82 | 1.45 | 61.33 | 5.93 | 6.87 | 23.45 | 42.57 |
| | 10 | | 32.05 | 0.95 | 42.57 | 3.68 | 5.78 | 24.26 | 55.09 |
| Swiss chard | | 10 | 25.1 | 1.07 b | 37.14 | 3.23 | 4.38 b | 22.27 | 20.97 |
| | | 1 | 18.2 | 1.64 a | 27.19 | 2.94 | 3.89 b | 24.76 | 21.00 |
| Sea beet | | 10 | 42.3 | 1.33 ab | 63.37 | 5.15 | 8.10 a | 25.25 | 47.13 |
| | | 1 | 26.6 | 1.06 b | 40.52 | 4.46 | 4.55 b | 22.47 | 50.53 |
| | 0 | 10 | 35.76 | 1.38 ab | 59.21 | 5.36 | 6.64 | 22.82 | 27.86 |
| | | 1 | 23.81 | 1.82 a | 39.46 | 4.52 | 4.76 | 21.91 | 30.97 |
| | 10 | 10 | 31.60 | 1.03 b | 41.30 | 3.02 | 5.85 | 24.70 | 40.24 |
| | | 1 | 20.96 | 0.89 b | 28.25 | 2.87 | 3.67 | 25.32 | 40.56 |
| ANOVA | | | | | | | | | |
| Plant species (PS) | | | *** | ns | *** | *** | *** | ns | *** |
| IO concentration | | | ns | *** | ** | *** | * | ns | ** |
| N-NO$_3$ concentration | | | *** | ns | *** | ns | *** | ns | ns |
| PS × IO | | | ns | ns | ns | ns | ns | ns | ns |
| PS × N-NO$_3$ | | | ns | ** | ns | ns | ** | ns | ns |
| IO × N-NO$_3$ | | | ns | ** | ns | ns | ns | ns | ns |
| PS × IO × N-NO$_3$ | | | ns | ** | ns | ns | * | ns | ns |

Means (*n* = 3) flanked by the same letter are not statistically different for *p* = 0.05 after Duncan's test. Significance level: *** $p \leq 0.001$; ** $p \leq 0.01$; * $p \leq 0.05$; ns = not significant.

The total uptake of Cl was not significantly affected by the plant species, salinity and N concentration (Table 4). On average, the total uptake of K (+61%); Ca (+55%); Mg (+53%); and Na (+133%) was higher in SB than in SC. Compared to non-salinized plants, those that were grown with saline nutrient solution absorbed less P (−40%); K (−29%); Ca (−40%); and Mg (−16%), but more Na (+37%). Moreover, the uptake of K (−32%, on average) and Mg (−44% in SB) was lower (−33%) in plants that were grown with a low N concentration, compared to the plants that were supplied with standard nutrient solution.

*3.2. Leaf Production, Succulence and Mineral Content in Different Harvests*

3.2.1. Swiss Chard

The leaf FW and area were significantly reduced by the high salinity (respectively, −29% and −27%) and low N level (respectively, −28% and −27%,) of the nutrient solution (Table 5; Figure 1, left). In contrast, the leaf DW/FW ratio was significantly higher in plants that were grown with saline water (+31.3%) or at a low N concentration (+17%) than in

the controls (Table 5). The leaf FW decreased in successive harvests, while the opposite tendency was observed for the leaf DW/FW ratio (Table 5; Figure 1, left). Salinity and N level did not affect leaf succulence, which was significantly lower in the leaves of the second and third cut (Table 5). On average, the use of saline water did not influence the leaf content of N, P, Mg, Fe, and Zn (Tables 6 and 7). In contrast, high salinity significantly reduced the content of K ($-24\%$); Ca ($-37\%$); Cu ($-16\%$); Mn ($-47\%$); and the K/Na molar ratio ($-52\%$), while increasing the content of Cl ($+30\%$) and Na ($+72\%$) (Tables 6 and 7).

**Table 5.** Leaf production (fresh weight, FW), dry weight/fresh weight ratio (DW/FW), leaf area and succulence of Swiss chard plants grown hydroponically with different concentrations of the sea salt Instant Ocean (IO; 0 and 10 g L$^{-1}$) and nitrogen (1 and 10 N-NO$_3$ mM) in the nutrient solution. The leaves were harvested trice during the experiment (see Table 1 for more detailed information on the experiment).

| Cut | IO (g L$^{-1}$) | N-NO$_3$ (mM) | Leaf Production (kg m$^{-2}$ FW) | Leaf DW/FW (%) | Leaf Area (m$^2$ m$^{-2}$) | Leaf Succulence (kg m$^{-2}$) |
|---|---|---|---|---|---|---|
| 1st cut | 0 | 10 | 4.87 | 3.88 | 13.71 | 0.374 |
| | | 1 | 3.32 | 4.79 | 8.48 | 0.446 |
| | 10 | 10 | 4.08 | 5.43 | 9.56 | 0.456 |
| | | 1 | 2.67 | 6.63 | 5.43 | 0.539 |
| 2nd cut | 0 | 10 | 2.96 | 4.32 | 10.41 | 0.240 |
| | | 1 | 1.77 | 6.23 | 9.59 | 0.185 |
| | 10 | 10 | 1.27 | 6.31 | 7.10 | 0.199 |
| | | 1 | 1.48 | 6.57 | 7.32 | 0.188 |
| 3rd cut | 0 | 10 | 3.23 | 4.80 | 15.27 | 0.223 |
| | | 1 | 2.55 | 6.06 | 9.39 | 0.277 |
| | 10 | 10 | 2.00 | 7.28 | 10.21 | 0.202 |
| | | 1 | 1.74 | 7.28 | 7.82 | 0.226 |
| MEAN EFFECTS | | | | | | |
| | 0 | | 3.12 a | 5.01 b | 10.93 a | 0.291 |
| | 10 | | 2.21 b | 6.58 a | 7.98 b | 0.302 |
| | | 10 | 3.07 a | 5.34 b | 11.00 a | 0.282 |
| | | 1 | 2.25 b | 6.26 a | 8.06 b | 0.310 |
| 1st cut | | | 3.73 a | 5.18 c | 9.23 ab | 0.454 a |
| 2nd cut | | | 1.87 b | 5.86 b | 8.58 b | 0.203 b |
| 3rd cut | | | 2.38 b | 6.35 a | 10.44 a | 0.232 b |
| | 0 | 10 | 3.69 a | 4.33 | 13.13 | 0.279 |
| | | 1 | 2.55 ab | 5.70 | 9.13 | 0.302 |
| | 10 | 10 | 2.45 b | 6.34 | 9.08 | 0.286 |
| | | 1 | 1.96 b | 6.83 | 6.99 | 0.317 |
| 1st cut | 0 | | 4.09 | 4.34 | 10.72 | 0.410 |
| | 10 | | 3.37 | 6.03 | 7.49 | 0.498 |
| 2nd cut | 0 | | 2.37 | 5.28 | 9.94 | 0.212 |
| | 10 | | 1.38 | 6.44 | 7.22 | 0.194 |
| 3rd cut | 0 | | 2.89 | 5.43 | 12.33 | 0.250 |
| | 10 | | 1.87 | 7.28 | 9.02 | 0.214 |
| 1st cut | | 10 | 4.47 a | 4.65 | 11.63 a | 0.415 |
| | | 1 | 2.99 b | 5.71 | 7.17 c | 0.492 |
| 2nd cut | | 10 | 2.12 bc | 5.32 | 8.76 abc | 0.220 |
| | | 1 | 1.63 c | 6.40 | 8.46 bc | 0.186 |
| 3rd cut | | 10 | 2.62 bc | 6.04 | 12.38 a | 0.212 |
| | | 1 | 2.14 bc | 6.67 | 8.49 bc | 0.251 |
| ANOVA | | | | | | |
| IO concentration | | | *** | *** | *** | ns |
| N-NO$_3$ concentration | | | *** | *** | *** | ns |
| Cut | | | *** | ** | * | *** |
| IO × N-NO$_3$ | | | * | ns | ns | ns |
| IO × Cut | | | ns | ns | ns | ns |
| N-NO$_3$ × Cut | | | * | ns | ** | ns |
| IO × N-NO$_3$ × Cut | | | ns | ns | ns | ns |

Means ($n = 3$) flanked by the same letter are not statistically different for $p = 0.05$ after Duncan's test. Significance level: *** $p \leq 0.001$; ** $p \leq 0.01$; * $p \leq 0.05$; ns = not significant.

**Table 6.** Leaf concentration of nitrogen (N); phosphorus (P); potassium (K); sodium (Na); calcium (Ca); magnesium (Mg); and chloride (Cl), and K/Na molar ratio in Swiss chard plants grown hydroponically with different concentrations of the sea salt Instant Ocean (IO; 0 and 10 g L$^{-1}$) and nitrate (NO$_3$; 1 and 10 mM) in the nutrient solution. The leaves were harvested trice during the experiment (see Table 1 for more detailed information in the experiment).

| Cut | IO (g L$^{-1}$) | N-NO$_3$ (mM) | N | P | K | Ca | Mg | Cl | Na | K/Na |
|---|---|---|---|---|---|---|---|---|---|---|
| | | | \multicolumn{8}{c}{(g kg$^{-1}$ DW)} | | | | | | | |
| 1st cut | 0 | 10 | 48.7 | 1.7 | 80.5 | 7.1 b | 6.4 | 31.6 | 26.8 | 1.76 |
| | | 1 | 39.6 | 1.2 | 70.5 | 8.0 a | 5.2 | 34.6 | 26.7 | 1.57 |
| | 10 | 10 | 49.6 | 1.7 | 83.7 | 4.5 de | 7.1 | 43.3 | 44.3 | 1.10 |
| | | 1 | 31.7 | 1.5 | 42.8 | 4.1 e | 3.5 | 44.5 | 31.0 | 0.82 |
| 2nd cut | 0 | 10 | 51.2 | 1.3 | 91.2 | 7.7 ab | 9.0 | 34.1 | 34.3 | 1.60 |
| | | 1 | 39.3 | 1.1 | 70.0 | 7.2 ab | 6.1 | 37.7 | 32.8 | 1.28 |
| | 10 | 10 | 55.4 | 2.0 | 84.5 | 5.4 c | 12.4 | 39.6 | 56.3 | 0.88 |
| | | 1 | 44.2 | 1.5 | 46.3 | 4.9 cd | 8.1 | 48.3 | 80.6 | 0.33 |
| 3rd cut | 0 | 10 | 47.7 | 0.9 | 72.9 | 7.9 a | 9.2 | 35.5 | 41.1 | 1.05 |
| | | 1 | 42.9 | 1.3 | 68.1 | 7.3 ab | 6.2 | 40.1 | 44.4 | 0.90 |
| | 10 | 10 | 47.3 | 0.9 | 44.2 | 4.4 de | 9.8 | 43.9 | 63.3 | 0.42 |
| | | 1 | 39.9 | 1.0 | 44.5 | 4.5 de | 6.8 | 58.6 | 78.8 | 0.33 |
| \multicolumn{11}{c}{MEAN EFFECTS} | | | | | | | | | | |
| | 0 | | 44.9 | 1.2 | 75.5 a | 7.5 a | 7.0 b | 35.6 b | 34.3 b | 1.36 a |
| | 10 | | 44.7 | 1.4 | 57.7 b | 4.7 b | 8.0 a | 46.4 a | 59.0 a | 0.65 b |
| | | 10 | 50.0 a | 1.4 | 76.2 a | 6.2 | 9.0 a | 38.0 b | 44.3 | 1.14 a |
| | | 1 | 39.6 b | 1.3 | 57.1 b | 6.0 | 6.0 b | 44.0 a | 49.0 | 0.87 b |
| 1st cut | | | 42.4 c | 1.5 a | 69.4 ab | 5.9 | 5.6 b | 38.5 b | 32.2 b | 1.31 a |
| 2nd cut | | | 47.5 a | 1.5 a | 73.0 a | 6.3 | 8.9 a | 39.9 ab | 51.0 a | 1.02 ab |
| 3rd cut | | | 44.4 b | 1.0 b | 57.4 b | 6.0 | 8.0 a | 44.5 a | 56.9 a | 0.68 b |
| | 0 | 10 | 49.2 a | 1.3 | 81.6 | 7.5 | 8.2 | 33.7 c | 40.8 | 1.47 |
| | | 1 | 40.6 b | 1.2 | 69.5 | 7.5 | 5.8 | 37.5 bc | 39.4 | 1.25 |
| | 10 | 10 | 50.8 a | 1.5 | 70.8 | 4.8 | 9.8 | 42.3 b | 42.3 | 0.80 |
| | | 1 | 38.6 c | 1.3 | 44.6 | 4.5 | 6.2 | 50.5 a | 37.5 | 0.50 |
| 1st cut | 0 | | 44.2 b | 1.5 | 75.5 | 7.5 a | 5.8 c | 33.1 | 26.7 b | 1.67 |
| | 10 | | 40.6 c | 1.6 | 63.3 | 4.3 c | 5.3 c | 43.9 | 37.6 b | 0.96 |
| 2nd cut | 0 | | 45.2 b | 1.2 | 80.6 | 7.4 a | 7.5 b | 35.9 | 33.5 b | 1.44 |
| | 10 | | 49.8 a | 1.7 | 65.4 | 5.2 b | 10.3 a | 43.9 | 68.4 a | 0.61 |
| 3rd cut | 0 | | 45.3 b | 1.1 | 70.5 | 7.6 a | 7.7 b | 37.8 | 42.8 b | 0.98 |
| | 10 | | 43.6 b | 1.0 | 44.4 | 4.5 bc | 8.3 b | 51.2 | 71.1 a | 0.38 |
| 1st cut | | 10 | 49.1 b | 1.7 a | 82.1 | 5.8 | 6.8 | 37.4 b | 35.5 | 1.43 |
| | | 1 | 35.6 d | 1.4 ab | 56.7 | 6.1 | 4.4 | 39.5 ab | 28.8 | 1.19 |
| 2nd cut | | 10 | 53.3 a | 1.6 a | 87.9 | 6.5 | 10.7 | 36.8 b | 45.3 | 1.24 |
| | | 1 | 41.7 c | 1.3 abc | 58.2 | 6.1 | 7.1 | 43.0 ab | 56.7 | 0.81 |
| 3rd cut | | 10 | 47.5 b | 0.9 c | 58.5 | 6.1 | 9.5 | 39.7 ab | 52.2 | 0.73 |
| | | 1 | 41.4 c | 1.1 bc | 56.3 | 5.9 | 6.5 | 49.4 a | 61.6 | 0.62 |
| \multicolumn{11}{c}{ANOVA} | | | | | | | | | | |
| IO concentration | | | ns | ns | ** | *** | ** | *** | *** | *** |
| N-NO$_3$ concentration | | | *** | ns | ** | ns | *** | *** | ns | ** |
| Cut | | | *** | *** | * | ns | *** | *** | *** | *** |
| IO × N-NO$_3$ | | | *** | ns | ns | ns | * | * | ns | ns |
| IO × Cut | | | *** | ns | ns | * | *** | ns | * | ns |
| N-NO$_3$ × Cut | | | *** | * | ns | ns | ns | * | ns | ns |
| IO × N-NO$_3$ × Cut | | | *** | ns | ns | * | ns | ns | ns | ns |

Means (*n* = 3) keyed by the same letter are not statistically different for *p* = 0.05 after Duncan's test. Significance level: *** $p \leq 0.001$; ** $p \leq 0.01$; * $p \leq 0.05$; ns = not significant.

**Table 7.** Leaf concentration of copper (Cu); manganese (Mn); iron (Fe); and zinc (Zn) in Swiss chard plants grown hydroponically with different concentrations of the sea salt Instant Ocean (IO; 0 and 10 g L$^{-1}$) and nitrogen (1 and 10 N-NO$_3$ mM) in the nutrient solution. The leaves were harvested trice during the experiment (see Table 1 for more detailed information in the experiment).

| Cut | IO (g L$^{-1}$) | N-NO$_3$ (mM) | Cu | Mn | Fe | Zn |
|---|---|---|---|---|---|---|
| | | | | (mg kg$^{-1}$ DW) | | |
| 1st cut | 0 | 10 | 13.0 cd | 137.0 | 157.0 | 67.0 |
| | | 1 | 15.0 bcd | 202.0 | 161.0 | 79.0 |
| | 10 | 10 | 13.0 cd | 106.0 | 138.0 | 65.0 |
| | | 1 | 11.0 d | 82.0 | 136.0 | 52.0 |
| 2nd cut | 0 | 10 | 22.0 ab | 151.0 | 317.0 | 82.0 |
| | | 1 | 22.0 ab | 274.0 | 376.0 | 97.0 |
| | 10 | 10 | 20.0 abc | 126.0 | 244.0 | 98.0 |
| | | 1 | 14.0 cd | 111.0 | 302.0 | 87.0 |
| 3rd cut | 0 | 10 | 23.0 a | 156.0 | 269.0 | 88.0 |
| | | 1 | 22.0 ab | 256.0 | 231.0 | 108.0 |
| | 10 | 10 | 18.0 abcd | 92.0 | 198.0 | 93.0 |
| | | 1 | 22.0 ab | 106.0 | 142.0 | 80.0 |
| MEAN EFFECTS | | | | | | |
| | 0 | | 19.5 a | 196.0 a | 251.8 | 86.8 |
| | 10 | | 16.3 b | 103.8 b | 193.3 | 79.2 |
| | | 10 | 18.2 | 128.0 b | 220.5 | 82.2 |
| | | 1 | 17.7 | 171.8 a | 224.7 | 83.8 |
| 1st cut | | | 13.0 b | 131.8 b | 148.0 | 65.8 b |
| 2nd cut | | | 19.5 a | 165.5 a | 309.8 | 91.0 a |
| 3rd cut | | | 21.3 a | 152.5 a | 210.0 | 92.3 a |
| | 0 | 10 | 19.3 | 148.0 | 247.7 | 79.0 |
| | | 1 | 19.7 | 244.0 | 256.0 | 94.7 |
| | 10 | 10 | 17.0 | 108.0 | 193.3 | 85.3 |
| | | 1 | 15.7 | 99.7 | 193.3 | 73.0 |
| 1st cut | 0 | | 14.0 | 169.5 | 159.0 | 73.0 |
| | 10 | | 12.0 | 94.0 | 137.0 | 58.5 |
| 2nd cut | 0 | | 22.0 | 212.5 | 346.5 | 89.5 |
| | 10 | | 17.0 | 118.5 | 273.0 | 92.5 |
| 3rd cut | 0 | | 22.5 | 206.0 | 250.0 | 98.0 |
| | 10 | | 20.0 | 99.0 | 170.0 | 86.5 |
| 1st cut | | 10 | 13.0 | 121.5 b | 147.5 | 66.0 |
| | | 1 | 13.0 | 142.0 b | 148.5 | 65.5 |
| 2nd cut | | 10 | 21.0 | 138.5 b | 280.5 | 90.0 |
| | | 1 | 18.0 | 192.5 a | 339.0 | 92.0 |
| 3rd cut | | 10 | 20.5 | 124.0 b | 233.5 | 90.5 |
| | | 1 | 22.0 | 181.0 a | 186.5 | 94.0 |
| ANOVA | | | | | | |
| IO concentration | | | ** | *** | ns | ns |
| N-NO$_3$ concentration | | | ns | *** | ns | ns |
| Cut | | | *** | *** | ns | *** |
| IO × N-NO$_3$ | | | ns | *** | ns | ns |
| IO × Cut | | | ns | ns | ns | ns |
| N-NO$_3$ × Cut | | | ns | * | ns | ns |
| IO × N-NO$_3$ × Cut | | | * | ns | ns | ns |

Means ($n$ = 3) keyed by the same letter are not statistically different for $p$ = 0.05 after Duncan's test. Significance level: *** $p \leq 0.001$; ** $p \leq 0.01$; * $p \leq 0.05$; ns = not significant.

Reducing the N level decreased the content of N −21%; K (−25%); and Mg (−33%), and the K/Na molar ratio (−23%), while increasing the content of Cl (+16%) and Mn (+34%) (Tables 6 and 7). The N concentration of the nutrient solution did not significantly influence the leaf content of P, Ca, Na, Cu, Fe, and Zn (Tables 6 and 7).

The leaf content of Mg, Cl, Na, Cu, Mn, and Zn tended to increase during the experiment, while an opposite trend was observed for nitric N, P, and the K/Na molar ratio (Tables 6 and 7). The leaf content of N was higher at the second harvest, while the content of K, Ca, and Fe did not differ significantly in different harvests (Tables 6 and 7).

The leaf NO$_3^-$ content, expressed as mg kg$^{-1}$ FW, ranged from 199 (second cut, 0IO-1N) to 2746 mg kg$^{-1}$ FW (third cut, 10IO-10N) (Figure 2, left). The content of NO$_3^-$

decreased (−85%) at the lower N level in the nutrient solution and increased (+42%) in plants that were grown in the 10IO-10N solution.

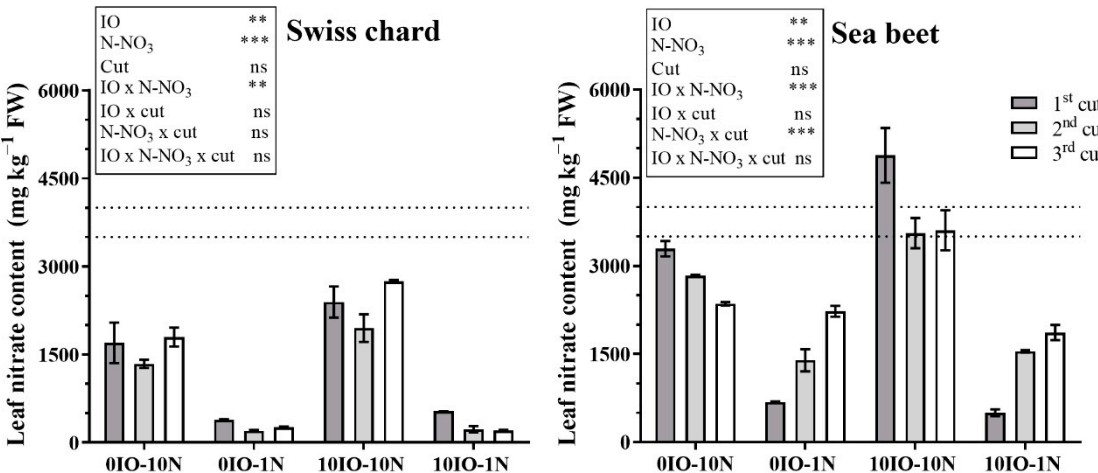

**Figure 2.** Leaf nitrate content of Swiss chard (left) and sea beet (right) grown hydroponically under greenhouse with different concentrations of the sea salt Instant Ocean (IO; 0 and 10 g L$^{-1}$) and nitrogen (1 and 10 N-NO$_3$ mM) in the nutrient solution (see Table 2 for abbreviations). Plants were harvested trice during the experiment. Dotted lines indicate the maximum levels of nitrates set by the Commission Regulation EU no. 1258/2011 for lettuce grown under greenhouse in spring–summer season and for spinach (4000 and 3500 mg kg$^{-1}$ FW, respectively). The mean values (*n* = 3) are reported with standard error. The results of ANOVA are reported inside the graph. Significance level: *** $p \leq 0.001$; ** $p \leq 0.01$; ns = not significant.

### 3.2.2. Sea Beet

The leaf FW and area were significantly reduced by the high salinity (respectively, −32% and −35%) and low N level (respectively, −24% and −18%) of the nutrient solution (Table 8; Figure 1, right). In contrast, the leaf DW/FW ratio was significantly higher in plants that were grown with saline water (+22%) or at a lower N concentration (+6%; Table 8).

Salinity and N level did not significantly affect leaf succulence, which was significantly lower in the leaves of the second and third cut than in the leaves of the first harvest (Table 8).

The leaf FW, area and succulence decreased during the experiment (Table 8; Figure 1, right), while the opposite tendency was observed for the leaf DW/FW ratio (Table 8).

On average, adding the sea salt to the nutrient solution did not affect the content of N, P, Mg, Fe, and Zn; in contrast, it significantly reduced the content of K (−18%); Ca (−28.2%); Cu (−18%); and Mn (−20%), and the K/Na (−48%) ratio, and increased the content of Cl (+22%) and Na (+55%) (Tables 9 and 10).

The reduction in N concentration in the nutrient solution significantly decreased the content of N (−20%); K (−20.5%); and Mg (−25%), and the K/Na (−42%) ratio, while increasing the content of Cl (+13%) and Na (+24%) (Table 9). The leaf content of P, Ca and trace elements did not significantly change in response to the nutrient solution N level (Tables 9 and 10).

The leaf content of P showed a tendency to increase in successive harvests, while an opposite trend was observed for Ca and Cu. Moreover, the N content was highest in the second cut. (Tables 9 and 10).

The leaf NO$_3$$^-$ content ranged from 500 (first cut, 10IO-1N) to 4880 mg kg$^{-1}$ FW (first cut, 10IO-10N; Figure 2, right). The reduction in N concentration in the nutrient solution resulted in a lower leaf NO$_3$$^-$ content (−60%). In contrast, the leaf NO$_3$$^-$ concentration increased (+22%) in salinized plants that were grown in the 10IO-10N solution (Figure 2, right).

**Table 8.** Leaf production (fresh weight, FW), dry weight/fresh weight ratio (DW/FW), leaf area and succulence of **sea beet** plants grown with hydroponically different concentrations of the sea salt Instant Ocean (IO; 0 and 10 g L$^{-1}$) and nitrogen (1 and 10 N-NO$_3$ mM) in the nutrient solution. The leaves were harvested trice during the experiment (see Table 1 for more detailed information on the experiment).

| Cut | IO (g L$^{-1}$) | N-NO$_3$ (mM) | Leaf Production (kg m$^{-2}$ FW) | Leaf DW/FW (%) | Leaf Area (m$^2$ m$^{-2}$) | Leaf Succulence (kg m$^{-2}$) |
|---|---|---|---|---|---|---|
| 1st cut | 0 | 10 | 6.32 | 5.43 | 14.84 a | 0.433 abc |
|  |  | 1 | 3.98 | 6.73 | 11.53 bc | 0.341 abc |
|  | 10 | 10 | 5.18 | 6.74 | 12.95 ab | 0.490 ab |
|  |  | 1 | 3.69 | 7.49 | 7.20 ef | 0.535 a |
| 2nd cut | 0 | 10 | 2.98 | 5.51 | 10.57 cd | 0.297 bc |
|  |  | 1 | 2.37 | 6.09 | 9.92 cd | 0.233 c |
|  | 10 | 10 | 1.38 | 7.14 | 6.23 f | 0.247 c |
|  |  | 1 | 1.47 | 7.07 | 3.44 g | 0.410 abc |
| 3rd cut | 0 | 10 | 3.81 | 6.84 | 11.41 bc | 0.297 bc |
|  |  | 1 | 3.02 | 7.29 | 9.01 de | 0.334 abc |
|  | 10 | 10 | 2.06 | 9.23 | 5.10 fg | 0.382 abc |
|  |  | 1 | 1.89 | 8.73 | 9.18 de | 0.240 c |
| MEAN EFFECTS |  |  |  |  |  |  |
|  | 0 |  | 3.84 a | 6.32 b | 11.25 a | 0.328 b |
|  | 10 |  | 2.61 b | 7.73 a | 7.35 b | 0.384 a |
|  |  | 10 | 3.62 a | 6.81 b | 10.18 a | 0.358 |
|  |  | 1 | 2.74 b | 7.23 a | 8.38 b | 0.349 |
| 1st cut |  |  | 4.79 a | 6.60 b | 11.63 a | 0.450 a |
| 2nd cut |  |  | 2.05 b | 6.45 b | 7.54 b | 0.297 b |
| 3rd cut |  |  | 2.70 b | 8.02 a | 8.67 b | 0.313 b |
|  | 0 | 10 | 4.37 a | 5.93 b | 12.27 | 0.342 |
|  |  | 1 | 3.12 ab | 6.70 ab | 10.15 | 0.303 |
|  | 10 | 10 | 2.88 ab | 7.70 a | 8.09 | 0.373 |
|  |  | 1 | 2.35 b | 7.76 a | 6.61 | 0.395 |
| 1st cut | 0 |  | 5.15 | 6.08 | 13.18 a | 0.387 |
|  | 10 |  | 4.44 | 7.12 | 10.07 b | 0.513 |
| 2nd cut | 0 |  | 2.67 | 5.80 | 10.24 b | 0.265 |
|  | 10 |  | 1.43 | 7.11 | 4.84 d | 0.329 |
| 3rd cut | 0 |  | 3.42 | 7.06 | 10.21 b | 0.316 |
|  | 10 |  | 1.98 | 8.98 | 7.14 c | 0.311 |
| 1st cut |  | 10 | 5.75 a | 6.09 b | 13.89 a | 0.462 |
|  |  | 1 | 3.83 b | 7.11 ab | 9.37 b | 0.438 |
| 2nd cut |  | 10 | 2.18 c | 6.32 b | 8.40 b | 0.272 |
|  |  | 1 | 1.92 c | 6.58 ab | 6.68 b | 0.321 |
| 3rd cut |  | 10 | 2.94 bc | 8.03 a | 8.25 b | 0.340 |
|  |  | 1 | 2.46 c | 8.01 a | 9.09 b | 0.287 |
| ANOVA |  |  |  |  |  |  |
| IO concentration |  |  | *** | *** | *** | * |
| N-NO$_3$ concentration |  |  | *** | * | *** | ns |
| Cut |  |  | *** | *** | *** | *** |
| IO × N-NO$_3$ |  |  | * | * | ns | ns |
| IO × Cut |  |  | ns | ns | * | ns |
| N-NO$_3$ × Cut |  |  | ** | * | *** | ns |
| IO × N-NO$_3$ × Cut |  |  | ns | ns | ** | ** |

Means ($n$ = 3) keyed by the same letter are not statistically different for $p$ = 0.05 after Duncan's test. Significance level: *** $p \le 0.001$; ** $p \le 0.01$; * $p \le 0.05$; ns = not significant.

**Table 9.** Leaf concentration of nitrogen (N); phosphorus (P); potassium (K); sodium (Na); calcium (Ca); magnesium (Mg); and chloride (Cl), and K/Na molar ratio of sea beet plants grown hydroponically with different concentrations of the sea salt Instant Ocean (IO; 0 and 10 g L$^{-1}$) and nitrogen (1 and 10 N-NO$_3$ mM) in the nutrient solution. The leaves were harvested trice during the experiment (see Table 1 for more detailed information in the experiment).

| Cut | IO (g L$^{-1}$) | N-NO$_3$ (mM) | N | P | K | Ca | Mg | Cl | Na | K/Na |
|---|---|---|---|---|---|---|---|---|---|---|
| | | | colspan: (g kg$^{-1}$ DW) | | | | | | | |
| 1st cut | 0 | 10 | 55.3 | 1.1 | 120.2 a | 8.9 | 7.6 | 27.1 | 47.9 | 1.47 a |
| | | 1 | 36.8 | 1.1 | 60.1 c | 8.9 | 5.5 | 32.0 | 74.5 | 0.48 ef |
| | 10 | 10 | 58.8 | 1.0 | 75.0 bc | 5.5 | 8.7 | 29.9 | 78.5 | 0.56 de |
| | | 1 | 36.9 | 1.0 | 62.4 c | 7.2 | 6.1 | 34.3 | 97.6 | 0.38 ef |
| 2nd cut | 0 | 10 | 61.3 | 1.5 | 93.0 b | 8.7 | 11.4 | 24.8 | 46.3 | 1.17 b |
| | | 1 | 47.9 | 1.4 | 70.4 bc | 7.4 | 7.4 | 24.5 | 77.6 | 0.53 def |
| | 10 | 10 | 59.6 | 1.3 | 64.0 c | 5.2 | 11.0 | 30.5 | 88.1 | 0.44 ef |
| | | 1 | 49.6 | 1.3 | 56.8 c | 5.5 | 8.7 | 37.3 | 91.5 | 0.37 f |
| 3rd cut | 0 | 10 | 51.8 | 1.5 | 67.7 bc | 6.0 | 8.4 | 29.9 | 52.4 | 0.75 c |
| | | 1 | 46.9 | 1.5 | 77.4 bc | 6.7 | 6.8 | 31.1 | 66.1 | 0.69 cd |
| | 10 | 10 | 53.8 | 1.3 | 75.4 bc | 4.8 | 9.5 | 35.3 | 93.9 | 0.47 ef |
| | | 1 | 54.2 | 1.6 | 67.3 bc | 5.3 | 7.7 | 41.3 | 99.2 | 0.40 ef |
| colspan: MEAN EFFECTS | | | | | | | | | | |
| | 0 | | 50.0 b | 1.3 | 81.9 a | 7.8 a | 7.9 | 28.5 b | 58.9 b | 0.85 a |
| | 10 | | 52.1 a | 1.2 | 66.8 b | 5.6 b | 8.6 | 34.8 a | 91.5 a | 0.44 b |
| | | 10 | 56.8 a | 1.3 | 82.6 a | 6.5 | 9.4 a | 29.6 b | 67.9 b | 0.81 a |
| | | 1 | 45.4 b | 1.3 | 65.7 b | 6.8 | 7.0 b | 33.4 a | 84.4 a | 0.47 b |
| 1st cut | | | 46.9 c | 1.0 b | 79.4 | 7.6 a | 7.0 b | 30.8 | 74.6 | 0.72 |
| 2nd cut | | | 54.6 a | 1.4 ab | 71.1 | 6.7 ab | 9.6 a | 29.3 | 75.9 | 0.63 |
| 3rd cut | | | 51.6 b | 1.5 a | 72.0 | 5.7 b | 8.1 b | 34.4 | 77.9 | 0.58 |
| | 0 | 10 | 56.1 | 1.4 | 93.6 | 7.9 | 9.2 | 27.3 | 48.9 d | 1.13 a |
| | | 1 | 43.8 | 1.3 | 69.3 | 7.7 | 6.5 | 29.2 | 72.7 c | 0.57 b |
| | 10 | 10 | 57.4 | 1.2 | 71.5 | 5.2 | 9.7 | 31.9 | 86.8 b | 0.49 b |
| | | 1 | 46.9 | 1.3 | 62.2 | 6.0 | 7.5 | 37.6 | 96.1 a | 0.38 b |
| 1st cut | 0 | | 46.0 | 1.1 | 90.2 | 8.9 | 6.5 | 29.6 | 61.2 | 0.98 |
| | 10 | | 47.8 | 1.0 | 68.7 | 6.4 | 7.4 | 32.1 | 88.1 | 0.47 |
| 2nd cut | 0 | | 54.6 | 1.5 | 81.7 | 8.1 | 9.4 | 24.7 | 62.0 | 0.85 |
| | 10 | | 54.6 | 1.3 | 60.4 | 5.3 | 9.8 | 33.9 | 89.8 | 0.40 |
| 3rd cut | 0 | | 49.3 | 1.5 | 72.6 | 6.4 | 7.6 | 30.5 | 59.3 | 0.72 |
| | 10 | | 54.0 | 1.5 | 71.4 | 5.1 | 8.6 | 38.3 | 96.6 | 0.44 |
| 1st cut | | 10 | 57.0 ab | 1.1 | 97.6 a | 7.2 | 8.1 | 28.5 | 63.2 | 1.02 |
| | | 1 | 36.8 d | 1.0 | 61.3 b | 8.0 | 5.8 | 33.2 | 86.1 | 0.43 |
| 2nd cut | | 10 | 60.5 a | 1.4 | 78.5 ab | 6.9 | 11.2 | 27.7 | 67.2 | 0.80 |
| | | 1 | 48.7 c | 1.3 | 63.6 b | 6.4 | 8.0 | 30.9 | 84.6 | 0.45 |
| 3rd cut | | 10 | 52.8 bc | 1.4 | 71.6 ab | 5.4 | 9.0 | 32.6 | 73.2 | 0.61 |
| | | 1 | 50.5 c | 1.5 | 72.4 ab | 6.0 | 7.3 | 36.2 | 82.7 | 0.54 |
| colspan: ANOVA | | | | | | | | | | |
| IO concentration | | | * | ns | ** | *** | ns | ** | *** | ** |
| N-NO$_3$ concentration | | | *** | ns | ** | ns | *** | * | *** | * |
| Cut | | | *** | ** | ns | * | *** | ns | ns | ns |
| IO × N-NO$_3$ | | | ns | ns | ns | ns | ns | ns | * | *** |
| IO × Cut | | | ns | ns | ns | ns | ns | ns | ns | ns |
| N-NO$_3$ × Cut | | | *** | ns | * | ns | ns | ns | ns | ns |
| IO × N-NO$_3$ × Cut | | | ns | ns | * | ns | ns | ns | ns | *** |

Means ($n = 3$) keyed by the same letter are not statistically different for $p = 0.05$ after Duncan's test. Significance level: *** $p \leq 0.001$; ** $p \leq 0.01$; * $p \leq 0.05$; ns = not significant.

**Table 10.** Leaf concentration of copper (Cu), manganese (Mn), iron (Fe), and zinc (Zn) in leaves of **sea beet** plants grown hydroponically with different concentrations of the sea salt Instant Ocean (IO; 0 and 10 g L$^{-1}$) and nitrogen (1 and 10 N-NO$_3$ mM) in the nutrient solution. The leaves were harvested trice during the experiment (see Table 1 for more detailed information in the experiment).

| Cut | IO (g L$^{-1}$) | N-NO$_3$ (mM) | Cu | Mn | Fe | Zn |
|---|---|---|---|---|---|---|
| | | | (mg kg$^{-1}$ DW) | | | |
| 1st cut | 0 | 10 | 24.0 | 215.0 | 199.0 | 96.0 |
| | | 1 | 19.0 | 149.0 | 174.0 | 66.0 |
| | 10 | 10 | 18.0 | 205.0 | 198.0 | 99.0 |
| | | 1 | 21.0 | 270.0 | 269.0 | 92.0 |
| 2nd cut | 0 | 10 | 20.0 | 196.0 | 253.0 | 71.0 |
| | | 1 | 15.0 | 139.0 | 260.0 | 71.0 |
| | 10 | 10 | 16.0 | 185.0 | 213.0 | 96.0 |
| | | 1 | 12.0 | 179.0 | 133.0 | 79.0 |
| 3rd cut | 0 | 10 | 11.0 | 159.0 | 116.0 | 87.0 |
| | | 1 | 7.0 | 119.0 | 177.0 | 60.0 |
| | 10 | 10 | 16.0 | 205.0 | 298.0 | 60.0 |
| | | 1 | 24.0 | 215.0 | 199.0 | 96.0 |
| MEAN EFFECTS | | | | | | |
| | 0 | | 18.5 a | 207.8 a | 189.6 | 82.4 |
| | 10 | | 15.2 b | 167.0 b | 220.0 | 75.3 |
| | | 10 | 16.0 | 177.2 | 204.8 | 72.8 b |
| | | 1 | 17.5 | 194.2 | 212.8 | 84.8 a |
| 1st cut | | | 20.8 a | 194.0 | 196.8 | 82.5 |
| 2nd cut | | | 18.0 a | 197.5 | 248.8 | 82.5 |
| 3rd cut | | | 11.5 b | 165.5 | 181.0 | 71.5 |
| | 0 | 10 | 18.3 | 218.7 a | 206.0 | 80.0 |
| | | 1 | 18.3 | 190.0 a | 189.3 | 84.7 |
| | 10 | 10 | 13.7 | 135.7 b | 203.7 | 65.7 |
| | | 1 | 16.7 | 198.3 a | 236.3 | 85.0 |
| 1st cut | 0 | | 23.0 | 211.0 | 207.5 | 82.5 |
| | 10 | | 18.5 | 177.0 | 186.0 | 82.5 |
| 2nd cut | 0 | | 20.5 | 233.0 | 261.0 | 81.5 |
| | 10 | | 15.5 | 162.0 | 236.5 | 83.5 |
| 3rd cut | 0 | | 11.5 | 169.0 | 124.5 | 83.0 |
| | 10 | | 11.5 | 162.0 | 237.5 | 60.0 |
| 1st cut | | 10 | 20.5 | 178.0 | 195.0 | 67.5 |
| | | 1 | 21.0 | 210.0 | 198.5 | 97.5 |
| 2nd cut | | 10 | 18.0 | 204.5 | 264.5 | 81.5 |
| | | 1 | 18.0 | 190.5 | 233.0 | 83.5 |
| 3rd cut | | 10 | 9.5 | 149.0 | 155.0 | 69.5 |
| | | 1 | 13.5 | 182.0 | 207.0 | 73.5 |
| ANOVA | | | | | | |
| IO concentration | | | * | * | ns | ns |
| N-NO$_3$ concentration | | | ns | ns | ns | * |
| Cut | | | *** | ns | ns | ns |
| IO × N-NO$_3$ | | | ns | * | ns | ns |
| IO × Cut | | | ns | ns | ns | ns |
| N-NO$_3$ × Cut | | | ns | ns | ns | ns |
| IO × N-NO$_3$ × Cut | | | ns | ns | ns | ns |

Means (*n* = 3) keyed by the same letter are not statistically different for *p* = 0.05 after Duncan's test. Significance level: *** *p* ≤ 0.001; * *p* ≤ 0.05; ns = not significant.

## 4. Discussion

### 4.1. Leaf Mineral Relations

During the experiment, no plant of both SC and SB showed evident symptoms of salt toxicity or nutrient deficiency (e.g., leaf chlorosis, necrosis or scorch). Leaf concentrations of macronutrients and trace elements (Tables 4, 5, 7 and 8) were invariably within the adequate ranges that were reported for many leafy vegetables [25], except for P, which was between 0.9 and 2.0 g kg$^{-1}$ DW with no important differences between SC and SB. The interaction among salinity, N level and cut was not significant for most of the measured quantities.

In both species, the N concentration of the nutrient solution reduced primarily the leaf NO$_3^-$ content (Figure 2). The higher leaf NO$_3^-$ content in SB than in SC, which was

observed in all the treatments (Figure 2), could be due to the greater N uptake (Table 4) and less efficient N assimilation of the wild species.

In most cases, in both species, the leaf mineral content was not affected by the composition of the nutrient solution, and it did not change during the growing period (Tables 4, 5, 7 and 8). However, the lower leaf concentration of Ca in salinized plants could be due to the antagonism with Na and Mg, which were dissolved in the nutrient solution at a concentration that was much higher than Ca. Calcium deficiency often occurs in plants that are exposed to a high Na concentration in the root zone, as found in *B. vulgaris* [26], in the halophyte *Portulaca oleracea* [27], and in many other plant species [28] A different result was observed in this work regarding the leaf Cu concentration, which increased in salinized plants of SC and decreased in SB (Tables 5 and 8).

In both species, reducing the N level in the nutrient solution resulted in a lower leaf content of K and Mg (Tables 4 and 7). A positive correlation between K and $NO_3^-$ uptake was found in several plant species [29]; however, in SC Na, but not K, this enhanced the translocation of $NO_3^-$ from roots to leaves [30].

Little differences, albeit significant, were observed among harvests regarding the leaf concentration of P, Ca, Mg, Cu, Mn, and Zn (Tables 4, 5, 7 and 8). In SC, the increase in leaf content of Mg, Cl, Na, Cu, Mn, and Zn at successive cuts could be ascribed to a higher nutrient uptake during the regrowth following the first and second cut, which resulted from the gradual growth of roots during the experiment. In SC plants that were grown hydroponically with standard nutrient solution or fish wastewater, the leaf content of Fe, Cu, Mn, and Zn remained constant throughout three consecutive harvests [6].

The plant response to NaCl salinity involves several molecular and physiological processes, including the regulation of the flux of $K^+$, $Na^+$ and $Cl^-$ for osmoregulation and ion homeostasis [31]. It is thought that the mechanisms of $Na^+$ and $K^+$ uptake and transport within the plant are basically the same in glycophytes and halophytes, but the latter are more tolerant to the ionic stress. Recently, Yolcu et al. [12] reviewed the different responses to salt stress in sea beet and the cultivated varieties of *B. vulgaris* when exposed to drought and salt stress. Among these responses, there are a better control of Na and Cl transport to young organs by shelding the old leaves, and an increase in leaf succulence in sea beet.

In this work, the salt-tolerant SC and the facultative halophyte SB showed an almost identical behavior when exposed to high salinity, both under optimal and sub-optimal N nutrition. Since, in both species, the use of salinized nutrient solution resulted in increased Cl and Na concentrations in both leaves and roots, and in a lower leaf K concentration and K/Na molar ratio, neither SC nor SB were able to limit the translocation of Na and Cl to the leaves. Both species were able to control the translocation of Cl to the leaves better than Na, as the ratio between the root content and total ion uptake over the growing period was, for Cl, approximately 24% in SC and 21% in SB, while for Na the ratios were 7% and 3% in SC and SB, respectively.

In both SC and SB, the higher leaf concentration of Na and Cl in plants that are grown with the lower N level in the nutrient solution, regardless of the salinity level (Tables 6 and 9), is in agreement with the reduction in Cl uptake in the glycophyte cucumber (*Cucumis sativus* L.) that is grown hydroponically with an increasing $NO_3^-$ concentration in the nutrient solution [32]. In contrast to our results, in *Suaeda salsa* L. (another species belonging to the Amaranthaceae) that was grown in sand culture, Na uptake increased with the N concentration in the culture solution [33], and $Cl^-$ had similar physico-chemical properties, resulting in a reciprocal antagonism regarding root uptake [34]. In the present work, the higher Cl and Na content in the leaves of the SC and SB plants that were grown with 1 mM of $N-NO_3^-$ solution was due to the reduction in leaf growth (Tables 5 and 8).

### 4.2. Crop Yield and Mineral Uptake

In the present work, all the plants of SC and SB grew well and healthy, albeit the growth rate depended on the species and the composition of the hydroponic nutrient solution. In general, both species showed a good adaptation to the hydroponic cultivation

in all the treatments. In the control, for instance, the crop yield exceeded 11 and 13 kg m$^{-2}$ in SC and SB, respectively (Table 3; Figure 1), which is remarkable considering the short growing period (less than two months). At the end of the experiment, no plant of either species showed symptoms of root or leaf rot, and the cultivation could probably have been extended for another two or four weeks, thus allowing for another harvest or two. The crop yield was evidently favored by the optimal greenhouse climatic conditions that typically occur during spring in the Mediterranean regions [35].

Neither plant species nor N nutrition affected the response to salinity, contrary to our expectations (see Introduction).

The crop yield of both species, also when grown in diluted seawater with 1 mM of N-NO$_3^-$, remained greater than the productions of leaves that were reported for *B. vulgaris* [16,36,37] and *Spinacia oleracea* [16], cultivated under greenhouse in hydroponics with standard nutrient solutions (the results of these studies are summarized in Table S5). Pantanella [19] reported that SB that was grown in saltwater (10 ppt) aquaponics yielded 2.6 kg m$^{-2}$ in four weeks.

Both salinity and N level influenced more plant water relations than dry biomass production. Indeed, the total crop water uptake was significantly reduced using saline water (Table 3). Contrasting results have been reported on the effect of salinity on the leaf water content of *Beta* species that are grown hydroponically. Sodium chloride levels up to 4.68 g L$^{-1}$ did not affect plant growth and leaf water content in SC [13]. High NaCl salinity (100 and 300 mM) reduced the leaf water content in sugar beet, while an opposite result was found in SB [38].

The growth inhibition in salinized SC and SB plants is consistent with previous findings in *Beta* species that were grown hydroponically. For instance, in SC that was grown in deep water culture with different NaCl concentrations in the nutrient solution, crop yield was not affected at 5 g L$^{-1}$ NaCl, while it was reduced by about 40% at 10 g L$^{-1}$ NaCl [9]. In SB, plant FW was reduced by roughly 50% in plants that were grown in gravel bed culture with 21.9 g L$^{-1}$ of NaCl in the nutrient solution [18], and by 25% in plants that were grown in pots irrigated with water containing 29.2 g L$^{-1}$ NaCl [39].

A lower yield of leafy vegetables that were grown hydroponically was found in plants grown with a reduced N supply (6 mM or less, compared to 12–14 mM; [40]. In wild Swiss chard (*Beta macrocarpa* Guss) that was cultured hydroponically with two levels of N concentration in the nutrient solution (0.5 and 2.5 mM), the total plant DW was markedly reduced at the lower N level [41]

In the present work, the growth reduction that was induced in SC and SB by high salinity and low N level in the culture solution was associated with a marked decrease in leaf area (Tables 5 and 8). The inhibition of leaf expansion is an adaptive response to salt stress, as it results in a lower transpiration rate. In sugar beet, moderate NaCl salinity limited the plant carbon assimilation, primarily due to a reduction in leaf area [42].

The production of fresh leaves changed in successive harvests in both SC and SB (Tables 5 and 8; Figure 1). The leaf biomass at the first cut was much greater than at successive harvests, and this was likely the result of the longer growing period (from transplanting) and greater cumulative solar radiation than in the following growth phases (Table 1). It is well known that plant growth is strongly correlated with the level of intercepted radiation. The response of leafy vegetables to the 'cut and come again' harvest depends on the plant species and growing conditions. A reduction in leaf production in successive harvests was found in basil [43] and in Swiss chard [6,9] that were grown in a floating system.

The total N uptake was greater in SB than in SC, while the difference in P uptake was not significant (Table 4). Nitrogen uptake was significantly reduced in plants that were grown with 1 mM of N-NO$_3^-$ in the nutrient solution with respect to the control, but it was not influenced by salinity, in contrast to P uptake, which significantly decreased in salinized plants (Table 4). These differences can be ascribed to the effects of salinity and N

level on plant growth, as both factors had no (P) or little (N) effects on the concentration of these elements in plants.

The total uptake of K, Ca, and Mg also decreased in plants that were grown with saline water or with a reduced N supply (Table 4), due to the reduction in dry biomass production.

### 4.3. Leaf Quality

In this work, leaf quality was assessed through the determination of leaf succulence (i.e., the moisture content to area ratio), the content of dry matter and $NO_3^-$, and the nutritional values that were associated with the concentration of mineral elements. Leaf succulence affects leaf texture, which is an important sensory attribute and thus may influence consumer acceptance [44,45]. On the other hand, leaf dry matter content is often positively correlated with post-harvest shelf life and tolerance to the typical operations of minimal processing [46].

Increased leaf succulence is a typical plant response to salinity [12,47]. For instance, leaf succulence increases with nutrient solution salinity in *Tetragonia tetragonioides* (Pallas) Kuntz grown in a floating system [48]. In this work, SC and SB showed similar values of leaf succulence, while the leaf DW/FW ratio was greater in SB. In both species, neither salinity nor N level significantly influenced leaf succulence, which instead significantly decreased in successive harvests, while the DW/FW ratio increased (Tables 5 and 8). Leal et al. [49] reported that, in spinach, the irrigation of brackish water with salinities ranging between 0.8 and 7.5 dS m$^{-1}$ increased leaf succulence in plants that were grown in soil, while the opposite result was found in plants that were cultivated in a floating system [49].

The leaves of SC and SB were a good source of K, Cl, and Mn, as EDI (%) was always above 15% for these elements (Table S6). A significant contribution to human diet was also observed in some treatments for Mg, Cu, and Fe (Table S6). In general, SB leaves would provide a higher amount of minerals to the human diet. On the other hand, the risk of excessive mineral intake that was associated with a serving dose of 100 g was negligible, as the calculated HRI was much lower than 100% for the considered elements, including Na (Table S6). To overcome the safe daily intake of Na, the daily consumption of leaves with the highest Na content that are found in this work (80.6 and 97.6 g kg$^{-1}$ DW, respectively, for SC and SB; Tables 6 and 9) should be greater than 351 g for SC and 230 g for SB.

In all the treatments, the leaf $NO_3^-$ level was lower in SC than in SB (Figure 2) because of a lower N uptake (Table 4) and leaf DW/FW (Tables 5 and 8). In SC, the $NO_3^-$ content that was detected was always below the threshold values (3500–4000 mg kg$^{-1}$ FW) fixed by the European Regulation 1169/2011 for spinach and lettuce that are grown under greenhouse conditions during the spring–summer period [24]. This regulation has fixed no $NO_3^-$ limits for SC and SB.

Excessive $NO_3^-$ accumulation in plant leaves results from an imbalance between its uptake and assimilation, it and depends on plant species and various environmental and agronomical factors [50]. For instance, higher $NO_3^-$ levels are generally found in plants that are grown under low irradiance and favored by abundant N fertilization in soil culture or a high N level in the nutrient solution in hydroponic cultivation [50,51]. The present study was conducted in spring, when irradiance and temperature were generally more favorable to N assimilation and leaf $NO_3^-$ accumulation was limited.

Sodium chloride salinity reduces the root uptake and leaf accumulation of $NO_3^-$ due to the antagonistic inhibition of $NO_3^-$ by $Cl^-$ [51]. In several species that were grown in water culture, the leaf $NO_3^-$ content was lower with 40 or 60 mM NaCl in the nutrient solution, as compared to NaCl-free solution [52]. Conversely, in SC that was grown hydroponically, Kaburagi et al. [30] found that Na, not K, increased plant $NO_3^-$ uptake by enhancing its transport in the xylem from roots to leaves. Nitrate uptake was also stimulated by NaCl in the halophyte *Suaeda physophora* Pall [53]. Increased $NO_3^-$ accumulation may contribute to leaf osmotic adjustment in salt-treated plants. Nitrate is known to serve as an osmoticum in leafy vegetables [50]. However, Kaburagi et al. [30]

observed that the $NO_3^-$ contribution to leaf osmotic potential was not affected by NaCl salinity in SC.

In the present work, in both SC and SB, the leaf $NO_3^-$ content that was expressed on an FW basis was higher in salinized plants than in non-salinized plants (Figure 2), as a consequence of the higher DW/FW ratio (Tables 5 and 8).

According to the Regulation (EU) No. 1169/2011 [24], the contribution of a serving food dose is considered significant if it provides at least 15% of RI (Table S1). In this work, the EDI (mg day$^{-1}$) that was calculated for a serving dose of 100 g of fresh leaves of both species was above 15% of RI for K, Cl, and Mn, regardless of cut time and the nutrient solution composition, except for K in SC leaves at the first harvest in the 10IO-1N treatment (Table S7). A significant contribution to RI was also observed for Mg in the 10IO-10N plants of SC at the second and third cut, and in the SB plants of the treatments 10IO-10N (all the cuts); 0IO-10N (second and third cut); and 10IO-1N (second and third cut; Table S7). The contribution to the RI of Cu was significant in the 10IO-1N plants of SC at the last harvest and in the 0IO-1N plants of SB at the first harvest. As regards Fe, EDI exceeded 15% of RI in the 0IO-1N plants of SC at the second cut, and in the 10IO-1N plants of SB at the third cut (Table S7).

## 5. Conclusions

Swiss chard and sea beet adapted well to the hydroponic culture with recurrent harvests, as they gave an abundant yield in short time. The response to the composition of the culture solution was almost identical in the two species, which grew rather well at high salinity, although growth was optimal under non-saline conditions. In both species, the use of diluted seawater reduced more plant water uptake than dry biomass production.

The leaves of both *Beta* species were a good source of potassium, chloride, and manganese for the human diet, while the risk of excessive sodium intake with a reasonable serving dose of fresh leaves was negligible. Sea beet accumulated more nitrates than Swiss chard. Nevertheless, in the most harvests, the leaf nitrate content is lower than the current limits that are fixed by the European Commission for spinach and lettuce that are grown under greenhouse conditions during the spring–summer period.

Overall, our findings indicate that both species could be successfully cultivated in hydroponics using diluted seawater.

**Supplementary Materials:** The following supporting information can be downloaded at: https://www.mdpi.com/article/10.3390/horticulturae8070638/s1, Table S1: Reference intake for an average adult, as reported in the Commission Regulation (EU) No 1169/2011, and tolerable upper intake level (UIL), safe and adequate intake (SAI), or acceptable daily intake (ADI) of mineral elements and nitrate ($NO_3$) set by the European Food Safety Authority * [54–57]. Table S2. Results of 4-way ANOVA with salinity and N-NO3 concentration of the nutrient solution, plant species and cut as variables. Table S3: Root concentration of nitrogen (N); phosphorus (P); potassium (K); calcium (Ca); magnesium (Mg); sodium (Na); and chloride (Cl) in Swiss chard and sea beet plants that are grown hydroponically with different concentrations of the sea salt Instant Ocean (IO; 0 and 10 g L$^{-1}$) and nitrate (N-NO$_3^-$; 1 and 10 mM) in the nutrient solution. The K/Na molar ratio is also shown. Table S4: Total root content of nitrogen (N); phosphorus (P); potassium (K); calcium (Ca); magnesium (Mg); sodium (Na); and chloride (Cl) in Swiss chard and sea beet plants that are grown hydroponically with different concentrations of the sea salt Instant Ocean (IO; 0 and 10 g L−1) and nitrate (N-NO3$^-$; 1 and 10 mM) in the nutrient solution. The K/Na molar ratio is also shown. Table S5: Leaf production of some vegetable species that are grown hydroponically under greenhouse conditions [16,36,37]. Table S6: Health risk index (HRI, %) associated with the consumption of 100 g FW of leaves of Swiss chard and sea beet plants that are grown hydroponically with different concentrations of the sea salt Instant Ocean (IO; 0 and 10 g L$^{-1}$) and nitrate (N-NO$_3^-$; 1 and 10 mM) in the nutrient solution (see Table 2 for abbreviations). The leaves were harvested trice during the experiment (see Table 1 for more information on the experiment and the test for the calculation of HRI). Table S7: Percentage of the reference intake for an average adult (as reported in the Commission Regulation (EU) No 1169/2011) associated with the consumption of 100 g of fresh leaves of Swiss chard or sea beet plants

that are grown hydroponically with different concentrations of the sea salt Instant Ocean (IO; 0 and 10 g L$^{-1}$) and nitrate (N-NO$_3$$^-$; 1 and 10 mM) in the nutrient solution (see Table 2 for abbreviations). The leaves were harvested trice during the experiment (see Table 1 for more information on the experiment and the text for the calculation of EDI).

**Author Contributions:** Conceptualization, M.P., A.P. and L.I.; Data curation, M.P.; Formal analysis, M.P., G.C., L.B., A.B. and J.F.F.-S.; Funding acquisition, A.P.; Investigation, M.P., G.C., A.P. and L.I.; Methodology, M.P. and A.P.; Project administration, A.P.; Resources, A.P.; Supervision, A.P.; Visualization, M.P., G.C., L.R., A.P. and L.I.; Writing—original draft, M.P. and A.P.; Writing—review and editing, M.P. and A.P. All authors have read and agreed to the published version of the manuscript.

**Funding:** This study was conducted within the framework of the EU-PRIMA (Section 2, 2018) project entitled "Self-sufficient Integrated MultiTrophic AquaPonic-SIMTAP" and financially supported by the Ministry of Education, University and Research (MIUR) of Italy.

**Institutional Review Board Statement:** Not applicable.

**Informed Consent Statement:** Not applicable.

**Data Availability Statement:** The data presented in this study are available within the article and supplementary material.

**Acknowledgments:** Fierro-Sañudo thanks the "Consejo Nacional de Ciencia y Tecnología" (CONACYT) for its support through the scholarship "Apoyo para Estancias Posdoctorales en el Extranjero Vinculadas a la Consolidación de Grupos de Investigación y Fortalecimiento del Posgrado Nacional 2019(1)".

**Conflicts of Interest:** The authors declare no conflict of interest.

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
