# Peer review of "Growth and Mineral Relations of Beta vulgaris var. cicla and Beta vulgaris ssp. maritima Cultivated Hydroponically with Diluted Seawater and Low Nitrogen Level in the Nutrient Solution"

_horticulturae, doi:10.3390/horticulturae8070638_

Round 1

Reviewer 1 Report

The problem considered in the article is relevant.

The literature review as a whole reflects the current state of the issue.

The material is presented clearly. The manuscript is presented in a well-structured form and is read with interest. The scheme of the experiment is described in detail.

The studies used modern physical and chemical methods.

 Unique data have been obtained that can become the basis for developing the theory and mechanisms of plant salt tolerance. The results are beautifully illustrated with 10 tables and 3

The authors analyzed 62 literary sources, of which only 37% are works of the last 5 years. It is recommended to more carefully study the scientific data on the problem over the past 5 years and use them when describing the results. This will certainly make the manuscript more coherent and interesting for readers.

There are no excessive self-citations.

The results are statistically significant.

The conclusions obtained are consistent with the experimental results. When interpreting the data, the authors compared their results with the available literature data. The conclusions are valid.

The structure, volume of the manuscript and the significance of scientific data correspond to the subject of the journal and will help to increase its rating.

I recommend accepting the article for publication after minor corrections, namely, additions to literary sources over the past 5 years.

Author Response

Dear Reviewer,

Thank you for your comments.

We have modified the references cited in the manuscript and now the 50% (27/54) are works of the last 5 years (>2018). Many older references (10/54) regard beet species and cannot be replaced. Based on a literature search on Scopus, there are no many recent works on the salinity response of Swiss chard and seabeet.

Reviewer 2 Report

Dear authors,

There are too many results (tables and figuers). Please, rearrange the results shortly. And there is no explain about experiment design (i.e. completed randaom designd and replication treatments). Pleaes, clearfy the experiment design. Discussion was too descriptive and redundants, please, rewrite concisely. Thus, I would like to inform you that this manuscript was a minor revison after review.

Kind regards,

Reviewer A.

Author Response

Dear Reviewer,

Thank you for your comments.

"There are too many results (tables and figuers). Please, rearrange the results shortly."

The results have been reduced. We deleted Figure 2 and the description of daily N uptake in the text, since the part of the Discussion about saltwater aquaponics was deleted too. Moreover, the results of the leaf content of organic and nitric nitrogen have been replaced by those of total nitrogen.

"And there is no explain about experiment design (i.e. completed randaom designd and replication treatments). Pleaes, clearfy the experiment design."

This information has been added (line 96-97 in the revised ms.).

"Discussion was too descriptive and redundants, please, rewrite concisely."

Paragraph 4.4 has been removed, and the discussion has been rewritten removing the redundant part.

Reviewer 3 Report

The article includes a large dataset and is well written, the results are properly explained and the discussion shows the importance of the results obtained.

I have only a few minor comments. Information is needed on the number of plants sampled and used for each measurement for each species and each treatment in the material and methods section.

I would recommend not separating the Results section by species, but to  merge 3.2. 1 and 3.2.2 and add species effects and interactions with other factors for all parameters analysed.  

Section 4.4. of the Discussion can be shortened. 

Author Response

Dear reviewer,

thank you for your comments.

"I have only a few minor comments. Information is needed on the number of plants sampled and used for each measurement for each species and each treatment in the material and methods section."

This information has been added (line 134 and 137 in the revised ms.).

"I would recommend not separating the Results section by species, but to  merge 3.2. 1 and 3.2.2 and add species effects and interactions with other factors for all parameters analysed. " 

The interaction of plant species with other factors was tested by 4-way nested ANOVA with salinity and N-NO3 concentration of the nutrient solution, plant species, and cut as variables. Since the interaction ‘salinity x N level x cut x species’ was not significant for most of the measured parameters, we decided to separately analyse the effect of cut in SC and SB, for the sake of clarity. A table with the results of the 4-way ANOVA has been added in the supplementary materials (Table S2). Anyway, we tested the effect of interaction of plant species by 3-way nested ANOVA with salinity and N concentration in the nutrient solution of the following parameters: yield, leaf DW, Root DW, Total DW, Root/leaf, Water uptake and on the total uptake of macronutrients (Table 3, 4).

"Section 4.4. of the Discussion can be shortened. "

The discussion has been considerably shortened.